mechanical engineering/mechanics

robotic disassembly, remanufacturing, robotics, active compliance, peg–hole disassembly, remote centre compliance (RCC)

**Author for correspondence:**
Yongjing Wang
e-mail: y.wang@bham.ac.uk

†These authors contributed equally to the study.

# Peg–hole disassembly using active compliance

Yongquan Zhang[1,2,†], Hong Lu[1,†], Duc Truong Pham[2], Yongjing Wang[2], Mo Qu[2], Joey Lim[2] and Shizhong Su[2]

[1]School of Mechanical and Electronic Engineering, Wuhan University of Technology, Wuhan 430070, People's Republic of China
[2]Department of Mechanical Engineering, School of Engineering, University of Birmingham, Edgbaston, Birmingham B15 2TT, UK

YZ, 0000-0003-4768-7279; HL, 0000-0003-3620-1659;
DTP, 0000-0003-3148-2404; YW, 0000-0002-9640-0871;
MQ, 0000-0002-2333-4412; JL, 0000-0002-4207-8098

When considered in two-dimensional space, a cylindrical peg being withdrawn from a clearance-fit hole can exhibit one of four contact states: no contact, one-point contact, two-point contact and line contact. Jamming and wedging can occur during the two-point contact. Effective control of the two-point contact region can significantly reduce resistance in peg–hole disassembly. In this paper, we explore generic peg–hole disassembly processes with compliance and identify the effects of key parameters including the degree of compliance, the location of the compliance centre and initial position errors. A quasi-static analysis of peg–hole disassembly has been performed to obtain the boundary conditions of the two-point contact region. The effects of key variables on the two-point contact region have been simulated. Finally, peg–hole disassemblies with different locations of compliance centre achieved using active compliance have been experimentally investigated. The proposed theoretical model can be implemented to predict the range and position of the two-point contact region from the perspective of peg–hole disassembly.

## 1. Introduction

Remanufacturing is the process of returning a used product to at least the original equipment manufacturer's performance specification from the customer's perspective and giving the resultant product a warranty that is at least equal to that of a newly manufactured equivalent [1]. Remanufacturing helps the environment as well as bringing economic and social benefits [2,3]. A critical step in remanufacturing is the disassembly of the returned product, which is normally manually executed and can be labour intensive due to its complexity [4]. Disassembly using robots can improve the efficiency of the process [5,6].

The removal of a cylindrical peg from a clearance-fit cylindrical hole is common in disassembly [7]. The operation can represent many industrial tasks [8], such as pulling a shaft out of a journal bearing. Although there have been many studies related to peg–hole assembly [9], little fundamental investigation has been conducted for disassembly. Position errors can increase the contact forces between peg and hole, and even cause them to be damaged. Two basic position errors are defined in peg–hole assembly: axial misalignment and angular misalignment [10].

Simunovic and Whitney analysed peg–hole assembly using a compliant manipulator and obtained geometric and force equilibrium conditions for overcoming position errors and informing successful insertion. Based on their theoretical and experimental results, the remote compliance centre (RCC) device was developed to improve the accuracy and efficiency of peg–hole insertion. Whitney also proved that the peg–hole insertion process was affected by the location of the compliance centre which should be at or near the tip of the peg [11–14]. Trong *et al*. [15] developed a dynamic model of peg–hole assembly using a compliant manipulator, in which key factors, including gravity, inertia, dry friction and insertion speed, were considered. Haskiya *et al*. [16] presented the geometrical and dynamical conditions enabling successful chamferless peg–hole insertion by correcting position errors.

The original RCC device consists of flexible mechanical components that deform slightly under load and thus can passively compensate for small initial lateral and angular errors [17]. An RCC device contains two parallel platforms connected by compliant shear pads. Once a configuration is determined, it is difficult to change compliance parameters, such as the lateral stiffness, angular stiffness and location of the compliance centre [18]. To overcome the inflexibility of the original RCC device, Lee *et al*. [19,20] proposed a variable remote centre compliance (VRCC) device, in which the compliance centre can be freely adjusted, to adapt to different peg–hole dimensions.

Active compliance using a closed-loop control system with external position, force and torque sensors has been developed to realize flexible assembly [21]. Active compliance relies on the accuracy of the sensors and the response speed of the control system. Wang *et al*. [22] compared the advantages and disadvantages of passive compliance and active compliance and they found that the latter can theoretically satisfy most application requirements. However, active compliance can exhibit a slow response. Tang *et al*. [23] used active compliance to explore the compensation trajectory from a three-point contact condition when the peg is outside the hole. They experimentally showed that misalignments corresponding to the three-point contact can effectively be eliminated by the proposed active method. Zhang *et al*. [24] proposed a fuzzy force control strategy to realize peg–hole assembly. Despite the active compliance approach being more expensive and having a limited response speed, it is an effective method to improve the reliability of assembly operation [25]. In addition, the compliance principle has also been applied in multiple-peg–multiple-hole assembly tasks to correct lateral and angular misalignments [26,27].

The focus of this paper is the quasi-static analysis of peg–hole disassembly using a compliance device and the effects of key variables including the degree of compliance, the location of the compliance centre and initial position errors. Such a fundamental analysis does not currently exist and is critical to a better understanding of one of the key tasks in disassembly.

The paper is organized as follows: §2 presents the definitions and assumptions adopted for the peg–hole disassembly analysis. §3 analyses the contact forces between peg and hole and derives the boundary conditions of the two-point contact region, showing how a compliant manipulator helps peg–hole disassembly. §4 analyses the two-point contact state to reveal the effects of key variables. §5 presents the experimental design and the results that confirm the proposed theoretical disassembly model.

# 2. Definitions and assumptions

The forces and moments acting on the peg–hole system, as well as its geometrical parameters, are defined in this section. In addition, assumptions about the initial conditions of the peg–hole system are stated.

## 2.1. Coordinate frame

Although peg–hole disassembly is a three-dimensional problem, it can be schematically illustrated and analysed in two dimensions [7]. A compliant manipulator provides lateral and angular compliance and can be modelled as a compliance centre, normally marked as ◑ and denoted by $O_C$, as shown in figure 1. The forces and moments on the peg can be represented by $F_x$, $F_z$ and $M$.

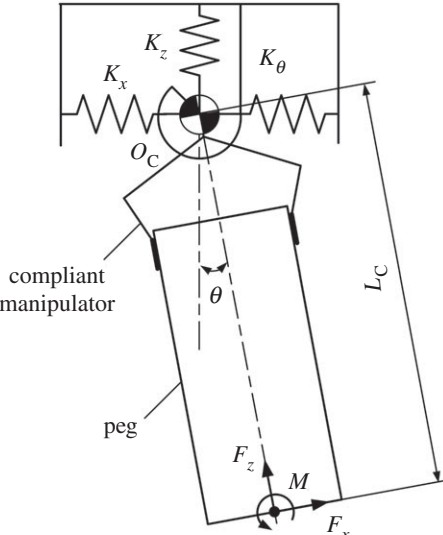

**Figure 1.** Definition of coordinate frame.

## 2.2. Initial conditions

Position errors including lateral and angular errors mainly consist of two parts: errors between the peg and the hole, and those between the compliant manipulator and the peg–hole system. Initially, as shown in figure 2, the peg and hole are assumed in the two-point contact, in which the peg is tilted by $\theta_0$. When the initial lateral error $\delta_0$ between the compliant manipulator and the peg–hole system is introduced, it also generates an angular error $\beta_0$.

## 2.3. Typical contact states during disassembly

Considering the typical mating geometries shown in figure 3, the peg–hole disassembly process can be divided into four main states: (a) no contact, (b) one-point contact, (c) two-point contact and (d) line contact. The no-contact state is ideal, but it rarely occurs. It will not be considered further in this paper. In general, the process starts with the two-point contact. When a peg is grasped by a compliant manipulator, initial lateral and angular errors drive the peg to shift and rotate, resulting in the two-point contact (figure 3c). The errors reduce as the extraction of the peg continues, and the process may transfer to the one-point contact (figure 3b) or line contact (figure 3d), depending on the compliance, the location of the compliance centre and the initial positioning errors. In a good extraction process, the peg would rotate to move from the two-point to one-point contact (figure 3b). The state would be maintained until the peg is totally extracted from the hole.

Jamming and wedging are well-known problems in peg–hole assembly [28]. Jamming is a condition in which the peg is not able to move due to improperly applied forces and moments. Wedging is a condition in which the peg is stuck at a position despite high insertion forces. These problems occur during the two-point contact state in peg–hole assembly, as well as disassembly. Reducing the two-point contact region, and controlling the position where two-point contact occurs, are useful strategies to avoid jamming and wedging [15].

# 3. Quasi-static analysis of peg–hole disassembly

Once the peg is grasped by a compliant manipulator, the peg would shift and rotate around the compliance centre in response to the initial position errors. The location of the compliance centre determines whether the contact state is one-point, two-point or line contact. In this section, the boundary conditions of each contact state, including geometrical and mechanical parameters, are derived.

## 3.1. Static model for different contact states

Assuming the peg and hole are initially in the two-point contact, the geometry of the peg and hole, and the forces and moments acting in the two-point contact stage, are illustrated in figure 4, in which $\delta_0$ and

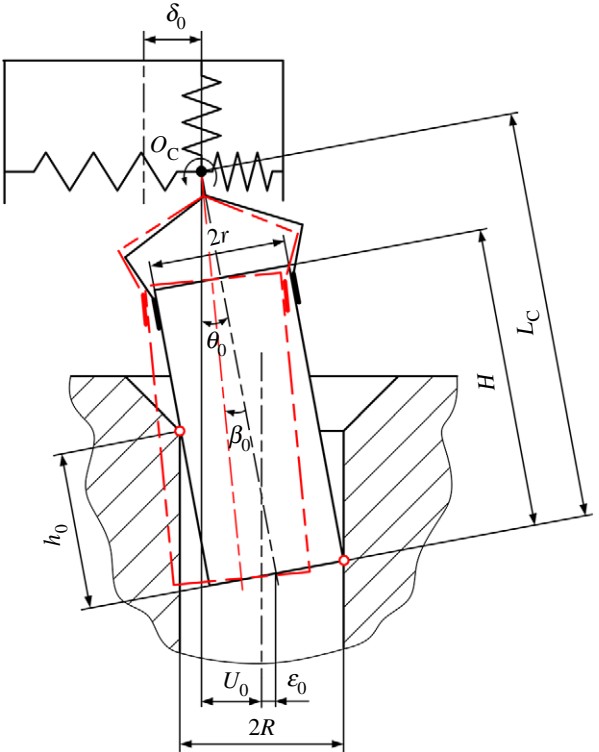

**Figure 2.** Definition of initial position.

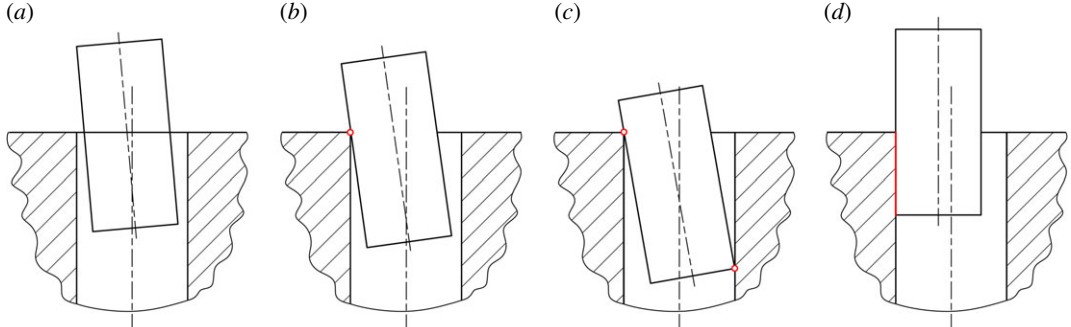

**Figure 3.** Typical states of the peg–hole disassembly process: (*a*) no contact, (*b*) one-point contact, (*c*) two-point contact and (*d*) line contact.

$\beta_0$ represent initial lateral and angular errors, respectively. The compliance centre and the tip of the peg are, respectively, located distances $U_0$ and $\varepsilon_0$ from the axis of the hole. The distances $U$ and $\varepsilon$ (figure 4) vary continuously during disassembly. The path of the compliance centre can be derived as

$$U - U_0 + (h\theta - \varepsilon_0) = L_C(\theta - \theta_0), \tag{3.1}$$

where $L_C$ is the distance between compliance centre and peg tip (figure 2) and $h$ is the peg extraction depth (figure 4).

According to geometrical constraints between the peg and the hole, the geometrical relations during two-point contact can be described as

$$R = \left(\frac{h}{2}\right)\sin\theta + r\cos\theta, \tag{3.2}$$

where $r$ and $R$ are the radii of the peg and the hole, respectively (figure 2).

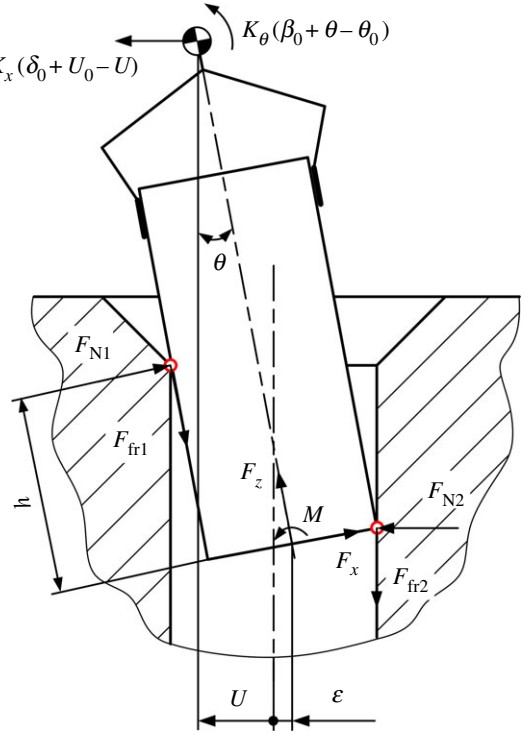

**Figure 4.** Geometry and forces during the two-point contact.

When the clearance between the peg and the hole is small, equation (3.2) can be simplified as

$$h\theta = 2cR, \tag{3.3}$$

where $c = (R - r)/R$ is defined as clearance ratio.

Based on the method proposed by Simunovic and Whitney, contact forces and supporting forces at and about the compliance centre can be re-expressed in the coordinate frame fixed at the tip of the peg.

$$\left.\begin{aligned}
F_x &= F_{N2} - F_{N1}\\
F_z &= \mu(F_{N1} + F_{N2})\\
M &= (h - \mu r)F_{N1} - \mu r F_{N2},
\end{aligned}\right\} \tag{3.4}$$

and

where $\mu$ is the coefficient of friction.

$$\left.\begin{aligned}
F_x &= -K_x(\delta_0 + U_0 - U)\\
M &= K_x L_C(\delta_0 + U_0 - U) + K_\theta(\beta_0 + \theta - \theta_0).
\end{aligned}\right\} \tag{3.5}$$

and

The boundary conditions of the two-point contact can be calculated if the contact force at point 2 (figure 4) is assumed to be $F_{N2} = 0$. The contact forces can then be calculated as

$$\left.\begin{aligned}
F_x &= -F_{N1}\\
F_z &= \mu F_{N1}\\
M &= (h - \mu r)F_{N1}.
\end{aligned}\right\} \tag{3.6}$$

and

Combining equation (3.1) through equation (3.6), the path of the compliance centre can be expressed as

$$\left.\begin{aligned}
U &= U_0 + \delta_0 - \frac{K_\theta(L_C\beta_0 + \delta_0 + 2cR - \varepsilon_0)}{K_x L_C(h - \mu r) - K_x L_C^2 + K_\theta}\\
\theta &= \theta_0 - \beta_0 - \frac{K_x(h - \mu r - L_C)(L_C\beta_0 + \delta_0 + 2cR - \varepsilon_0)}{K_x L_C(h - \mu r) - K_x L_C^2 + K_\theta}.
\end{aligned}\right\} \tag{3.7}$$

and

Substituting geometrical constraints (equation (3.2) and equation (3.3)) into equation (3.7), a quadratic equation in extraction depth $h$ for the start and endpoints of the two-point contact region can be derived as

$$\alpha h^2 + \beta h + \gamma = 0, \tag{3.8}$$

where,

$$\left.\begin{array}{l} \alpha = A + K_x L_C B \\ \beta = (E - \mu r K_x L_C)B - (L_C + \mu r)A - 2cR K_x L_C \\ \gamma = 2cR(\mu r K_x L_C - E) \\ A = K_x(L_C \beta_0 + \delta_0 + 2cR - \varepsilon_0) \\ B = \theta_0 - \beta_0 \\ E = K_\theta - K_x L_C^2, \end{array}\right\}$$

and

The solutions of equation (3.8) are

$$\left.\begin{array}{l} h_2 = \dfrac{-\beta + \sqrt{\beta^2 - 4\alpha\gamma}}{2\alpha} \\[2mm] h'_2 = \dfrac{-\beta - \sqrt{\beta^2 - 4\alpha\gamma}}{2\alpha}. \end{array}\right\} \tag{3.9}$$

and

In the calculation results, the solutions $h_2$ and $h'_2$ represent the start and the end of the two-point contact region, respectively. $h'_2 < h_0$ indicates that the peg and hole would be in the two-point contact when the peg is grasped by the compliant manipulator. $h_0 < h'_2 < h_2$ indicates that the peg and hole are initially in the one-point contact, and there is at least one transformation between the one-point contact state and the two-point contact state. If equation (3.8) has no solution, the two-point contact cannot occur during disassembly. The height and position of the two-point contact region depend not only on the geometrical parameters of the peg–hole system, but also on the location of compliance centre, initial position errors and the degree of compliance.

The geometrical parameters and the forces on the peg during the one-point contact are shown in figure 5. The geometric constraint in this state can be described as

$$U = L_C \theta - h\theta + cR. \tag{3.10}$$

The contact forces can be expressed in a coordinate frame fixed to the tip, as shown in equation (3.5). Combining equation (3.5) and equation (3.6) to represent the support forces acting at the compliance centre yields the equations for $\theta$ and $U$ for the one-point contact.

$$\left.\begin{array}{l} \theta = \dfrac{N_2 - K_x N_1(h - \mu r - L_C)}{K_x(h - L_C)(h - \mu r - L_C) - K_\theta} \\[3mm] U = \delta_0 + U_0 - \dfrac{N_2 + K_\theta \theta}{K_x(h - \mu r - L_C)}, \end{array}\right\} \tag{3.11}$$

and

where

$$\begin{array}{l} N_1 = \delta_0 + U_0 - cR \\ N_2 = K_\theta(\beta_0 - \theta_0). \end{array}$$

## 3.2. Forces acting during disassembly

This section discusses the extraction forces in the two-point contact and one-point contact states. Rearranging equation (3.4) yields the extraction force $F_z$ during the two-point contact.

$$F_z = \frac{M}{\lambda r} - \frac{\mu}{\lambda}(1 - \lambda)F_x, \tag{3.12}$$

where $\lambda = h/(2r\mu)$.

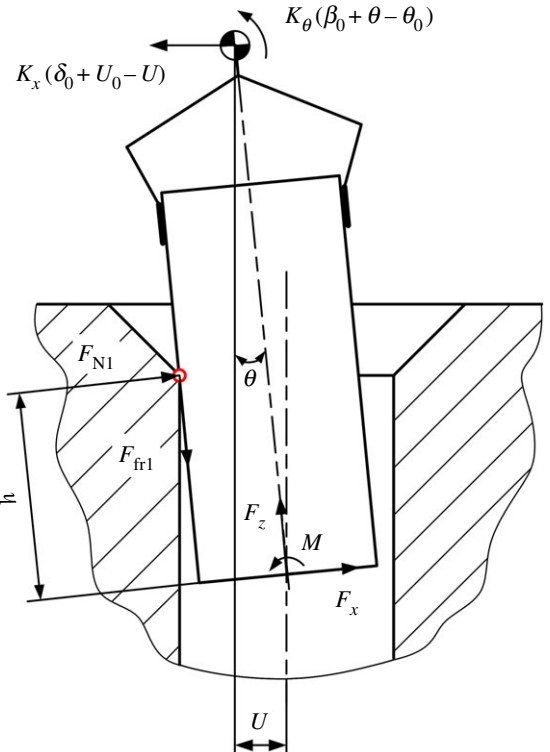

**Figure 5.** Geometry and forces during the one-point contact.

Substituting equation (3.1) and equation (3.3) into equation (3.5) yields $F_x$ and $M$ for the two-point contact state.

and
$$\left.\begin{array}{l} F_x = - K_x(\delta_0 - \varepsilon_0 + L_C\theta_0 + cD - cDL_C/h) \\ M = K_xL_C(\delta_0 - \varepsilon_0 + L_C\theta_0 + cD) - cDK_xL_C^2/h + K_\theta(\beta_0 - \theta_0) + cDK_\theta/h \end{array}\right\}, \quad (3.13)$$

where $D = 2R$ is the diameter of the hole.

Then, the extraction force in the two-point contact state can be calculated by substituting equation (3.13) into equation (3.12).

To obtain the extraction force $F_z$ during the one-point contact, substituting equation (3.11) into equation (3.6) yields

$$F_z = \frac{\mu K_x K_\theta(\beta_0 - \theta_0 + \theta)}{K_x(h - \mu r - L_C)}. \quad (3.14)$$

# 4. Key factors and their effects

There is a greater risk of the extraction process failing in the two-point contact region especially if it is near the mouth of the hole. Two-point contact also increases the extraction force because of the lateral and angular errors. In this section, the most effective parameters to reduce the two-point contact region and their impact on its location are identified. To illustrate the effects of the variables on the two-point contact region, the following parameters' values were used.

## 4.1. Location of compliance centre

The location of the compliance centre at an arbitrary point along the peg's axis is an important design parameter. When the peg is grasped by a compliant manipulator, the peg ought to shift and rotate due to initial position errors between the compliant manipulator and the peg–hole system. Assuming that the rectangle with solid lines, as shown in figure 6a,b, represents the initial position of the peg without initial angular error, the peg would shift and rotate around the compliance centre due to the

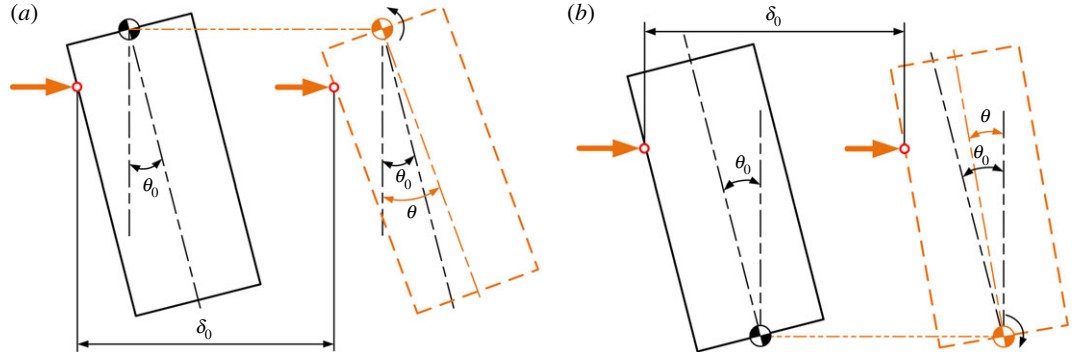

**Figure 6.** Response of the peg to the different locations of the compliance centre: (*a*) compliance centre at base of peg, $L_C = 50$ mm; (*b*) compliance centre at tip of peg, $L_C = 0$ mm.

initial lateral error. The grasping of the peg causes contact between the peg and the mouth of the hole and thus a lateral force to shift the peg to a distance $\delta_0$. The force and distance depend on the extraction depth.

As shown in figure 6*a*, if the compliance centre is located far from the peg tip, the peg would shift and rotate anticlockwise around the compliance centre due to the lateral force. By taking into account the geometrical constraints of the hole, the shifting and rotation of the peg would transfer to elastic energy stored in compliant manipulator. The peg would shift and rotate clockwise around the compliance centre when it is placed at the tip of the peg, as shown in figure 6*b*, in which case the peg and the hole would be in one-point or line contact once the peg is grasped. These characteristics can help reduce the size of the two-point contact region during disassembly.

Based on equation (3.9), the boundary conditions of the two-point contact region are illustrated in figure 7. It can be seen that the two-point contact region is reduced significantly when the compliance centre is at the tip of the peg. Besides, the two-point contact region moves to the mouth of the hole with a decrease in $L_C$.

## 4.2. Initial position errors

The initial position errors, including the lateral and angular errors between the compliant manipulator and the peg–hole system, can also influence the two-point contact region. When the compliance centre is located far from the tip as shown in figure 6*a* and the lateral error $\delta_0$ is large, the anticlockwise rotation of the peg can be very high. If $\delta_0$ is large enough, the peg and the hole may remain in the two-point contact throughout the disassembly process. However, if the compliance centre is at the tip of the peg, the peg rotates clockwise during disassembly. This would transfer the peg and hole from the two-point contact to the one-point or line contact.

The effects of the initial lateral error, for different locations of compliance centre, on the two-point contact region are illustrated in figure 8. In this case, it can be seen that if the compliance centre is near the tip of the peg, the two-point contact region reduces with an increase in lateral error. However, an increase in lateral error has little impact on the two-point contact region when the compliance centre is far from the tip of the peg.

The effects of the initial angular error on the disassembly process depend not only on its magnitude but also on its direction. As shown in figures 9 and 10, if the initial angular error is opposite to the rotation of the peg, it helps reduce the two-point contact region when the compliance centre is far from the peg tip. Otherwise, it increases the two-point contact region. The effects of the initial angular error are similar when the compliance centre is at the tip of the peg.

## 4.3. Stiffness

The compliant manipulator provides both lateral and angular compliance defined by the lateral and rotational stiffness, respectively. When the angular stiffness is low, as shown in figure 11*a*, rotation is the dominant movement. If the compliance centre is far from the tip, this would result in an increase in the size of the two-point contact region. On the contrary, if the compliance centre is located near the tip of the peg, the peg and hole could move from the two-point contact to the one-point contact, leading to a reduction in the size of the two-point contact region.

From figure 12*a,b*, it can be surmised that the two-point contact region reduces to some extent with an increase in lateral stiffness wherever the compliance centre is located. Increasing the lateral stiffness

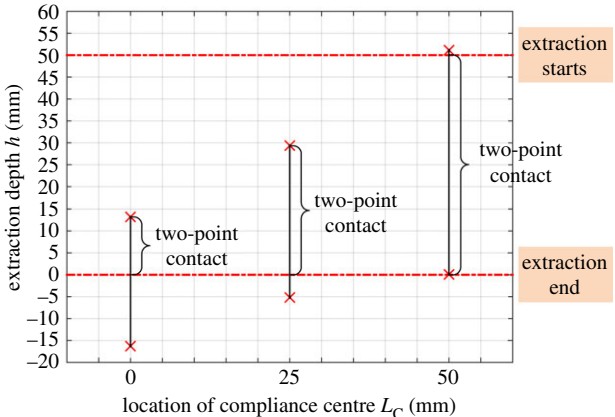

**Figure 7.** Dependence of the two-point contact region on the location of the compliance centre with $\delta_0 = 2\,\text{mm}$, $\beta_0 = 0\,\text{rad}$, $K_x = 4\,\text{N mm}^{-1}$, $K_\theta = 30\,\text{Nmm rad}^{-1}$.

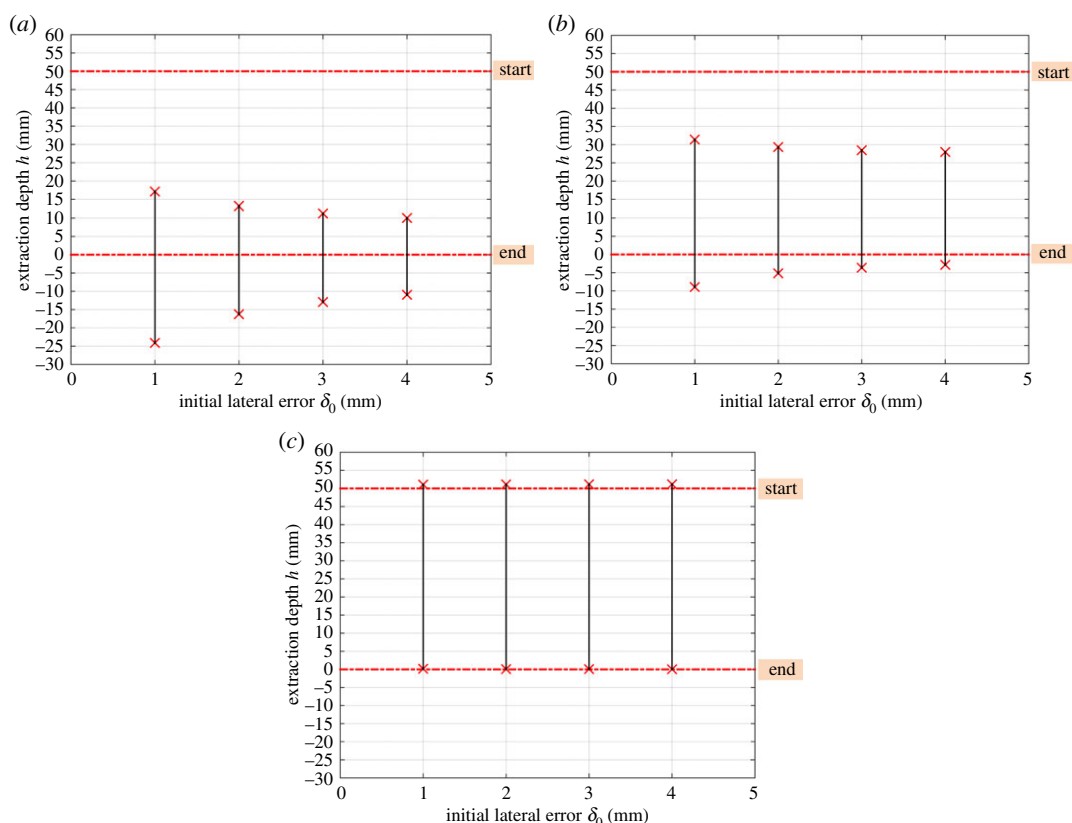

**Figure 8.** Dependence of the two-point contact region on the initial lateral error: (*a*) $L_C = 0\,\text{mm}$; (*b*) $L_C = 25\,\text{mm}$ and (*c*) $L_C = 50\,\text{mm}$ with $\beta_0 = 0\,\text{rad}$, $K_x = 4\,\text{N mm}^{-1}$, $K_\theta = 30\,\text{Nmm rad}^{-1}$.

restrains the rotation of the peg for the same position errors, thereby reducing the two-point contact region. Also, the two-point contact region reduces with a decrease in angular stiffness when the compliance centre is at the tip of peg, which benefits the disassembly process. However, the stiffness has little impact on the position of the two-point contact region in this case. In sum, the sensitivity of the two-point contact region to stiffness depends not only on the ratio of lateral stiffness and angular stiffness, but also on their absolute values.

## 4.4. Discussion

Taking into account geometrical constraints, the principal peg–hole disassembly processes are summarized in this section. The key factors are the location of the compliance centre, initial position

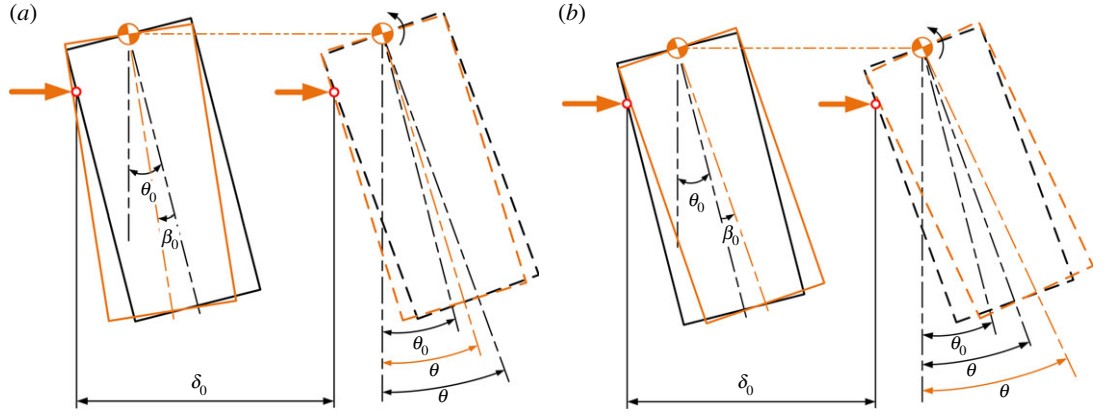

**Figure 9.** ($a,b$) Response of the peg to the different angular errors: $L_C = 50$ mm.

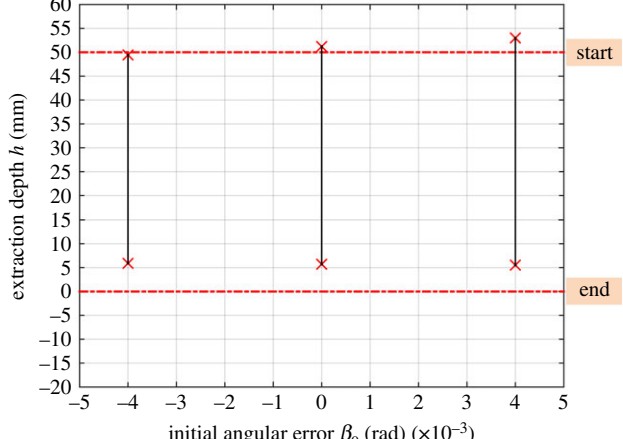

**Figure 10.** Dependence of the two-point contact region on the initial angular error: $L_C = 50$ mm, $\delta_0 = 1$ mm, $K_x = 4$ N mm$^{-1}$, $K_C = 2000$ Nmm rad$^{-1}$.

errors and stiffness. The following analysis is based on the assumption that the peg and hole are initially in the two-point contact.

If the compliance centre is located far from the peg tip, as shown in figure 13$a$, an initial lateral error will cause a lateral force acting on the compliance centre. Owing to geometrical constraints, the peg will rotate until the two-point contact occurs. With an increase in extraction distance, the lateral error reduces and the state transforms to the one-point contact or line contact until the peg is out of the hole. As a result, the disassembly process can be performed successfully with small lateral errors in this situation.

If the compliance centre is located near the tip of the peg, as shown in figure 13$b$, with the lateral force acting on the peg, the peg gradually rights itself as it is drawn out of the hole and it also shifts and rotates because of misalignments and geometrical constraints. However, due to the location of the compliance centre, the peg gradually rights itself as it is drawn out of the hole and there is no two-point contact region near the mouth of the hole. The peg and hole transfer from the initial two-point contact to the one-point contact, and this state remains until the peg is totally extracted. Clearly, this is a more desirable peg–hole disassembly process.

An extraction process may also start with the one-point contact, as shown in figure 13$c$. With an increase in extraction distance, the peg could enter the two-point contact region due to a large angular misalignment. If the angular error is small, the contact state may change to the one-point contact when the tip of the peg is near the mouth of the hole. In other cases, as shown in figure 13$d$, the two-point contact state may be maintained.

If the ratios of the relevant parameters are incorrect, an undesirable motion may be seen, as can be seen in figure 14. As the two-point contact is a state where jamming and wedging may occur, avoiding this contact type or keeping it in the region where the tip of the peg is far from the edge of hole can reduce the risk of failure in peg–hole separation.

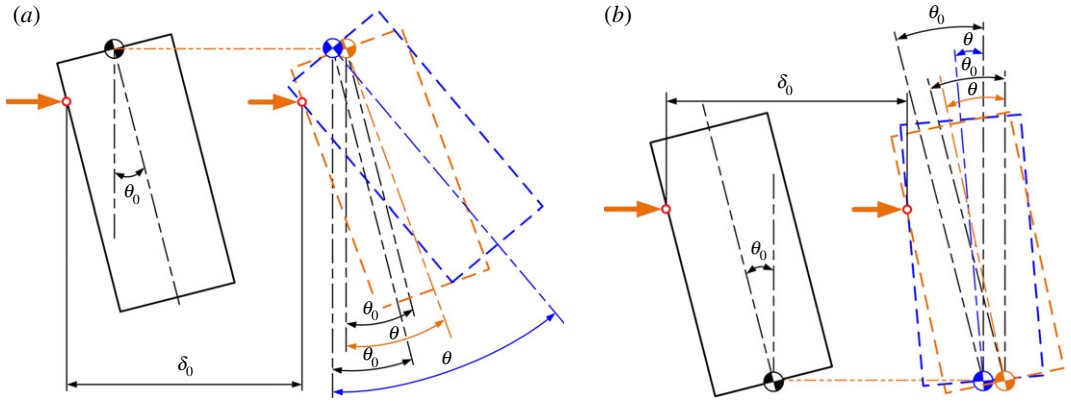

**Figure 11.** Response of peg to different proportions of lateral and angular stiffness: (*a*) $L_C = 50$ mm; (*b*) $L_C = 0$ mm.

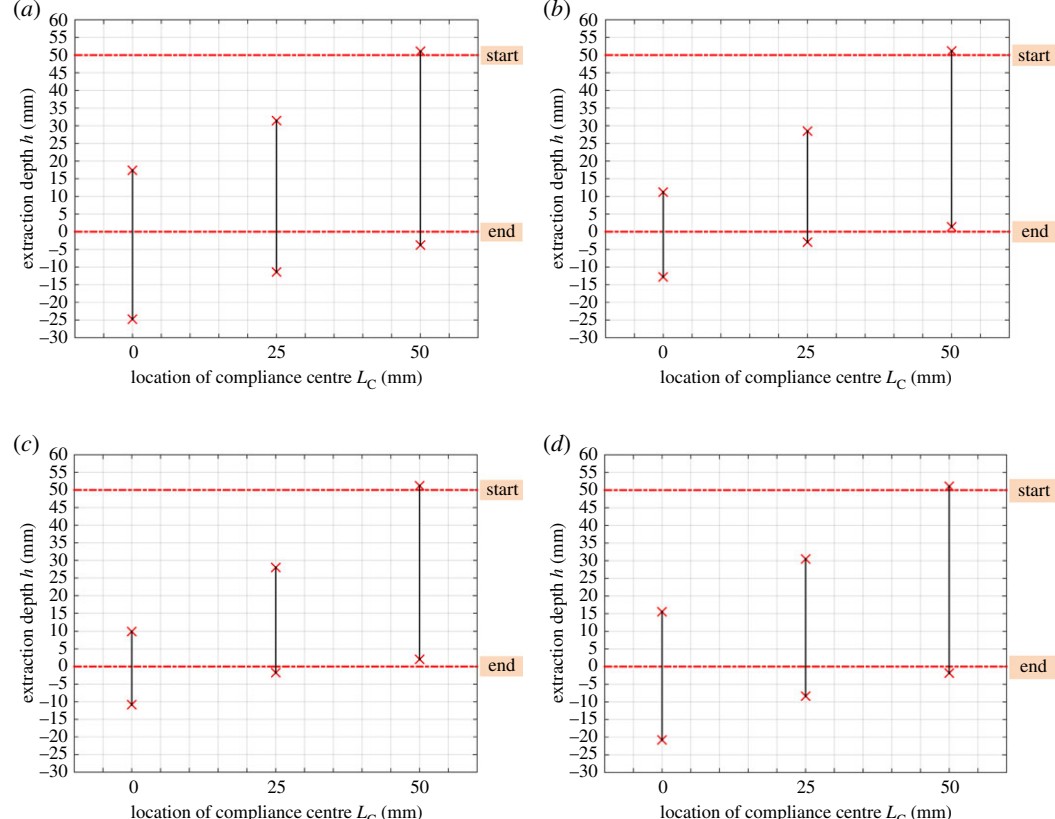

**Figure 12.** Dependence of the two-point contact region on the structural parameters: (*a*) $K_x = 2$ N mm$^{-1}$, $K_\theta = 30$ Nmm rad$^{-1}$; (*b*) $K_x = 6$ N mm$^{-1}$, $K_\theta = 30$ Nmm rad$^{-1}$; (*c*) $K_x = 4$ N mm$^{-1}$, $K_\theta = 15$ Nmm rad$^{-1}$; (*d*) $K_x = 4$ N mm$^{-1}$, $K_\theta = 45$ Nmm rad$^{-1}$ with $\delta_0 = 2$ mm, $\beta_0 = 0$ rad.

# 5. Experimental investigation

The effects of a compliant manipulator with different locations of compliance centre on the two-point contact region have been experimentally validated. The manipulator used was a KUKA LBR robot [29].

## 5.1. Experiment design

The experimental set-up is shown in figure 15. A KUKA robot (LBR iiwa 14 R820 [29]) and an external 6-DOF F/T sensor (ATI Theta F/T 20769 [30]) were used to perform the peg–hole disassembly process. The active compliance facility of the robot was employed to change the location of the compliance centre.

(a)

(b)

(c)

(d)

**Figure 13.** Comparison of peg movements for different locations of compliance centre (*a,b*), and different initial conditions (*c,d*).

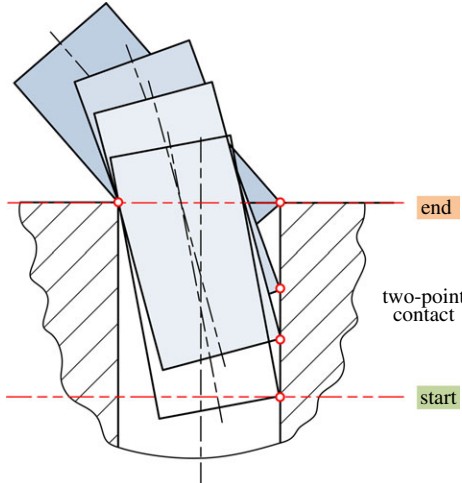

**Figure 14.** Ultimate part motions with respect to peg–hole disassembly.

Although the robot has in-built force sensors for compliance control, the external F/T sensor was used. The peg was fixed to the media flange of the robot arm, while the hole block was located on the F/T sensor. The material of the peg and hole was 45 steel, and the peg and hole were hardened and ground to minimize the effects of profile errors. The pose repeatability of the robot was ± 0.1 mm, and the resolution of the force of F/T sensor in the *z*-axis direction was 1 N. The experiment was repeated at least three times; the consistency of the results proved that the effects of the repeatability of the robots were negligible.

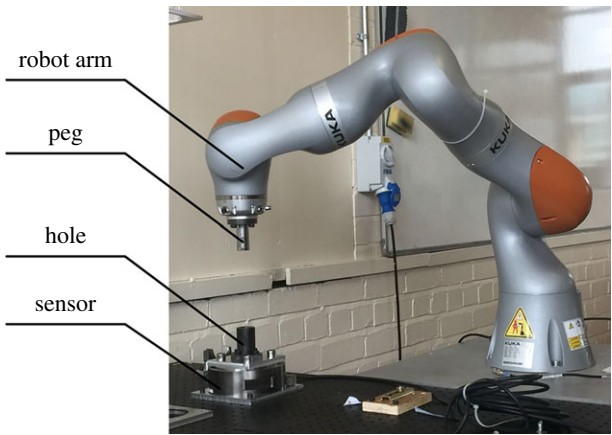

**Figure 15.** Experimental set-up of robotic disassembly.

**Table 1.** Parameters for peg–hole system.

| coefficient of friction | $\mu = 0.1$ | |
|---|---|---|
| geometrical parameters | peg mass | $m = 0.4\,\mathrm{kg}$ |
| | peg radius | $r = 12.44\,\mathrm{mm}$ |
| | hole radius | $R = 12.53\,\mathrm{mm}$ |
| | peg length | $L = 50\,\mathrm{mm}$ |
| | clearance ratio | $c = 0.0072$ |
| initial position/angle | depth | $h_0 = 50\,\mathrm{mm}$ |
| | angle | $\theta_0 = 0.0036\,\mathrm{rad}$ |

Active compliance was used to simulate compliant movements of a tool centre point (TCP). The peg could be rotated about the TCP without changing the position of the latter. Several TCPs were configured to simulate the different locations of the compliance centre. Additionally, lateral stiffness and angular stiffness can be programmed [29].

The greatest challenge in the experiment was to keep the same initial conditions in every test. To achieve this, the robot was programmed to move to the same start position before beginning to pull the peg out of the hole. The start position was chosen to induce the desired contact forces on the peg. It was found that the position repeatability of the robot (±0.1 mm) was sufficient to ensure the initial contact forces were within the required tolerance (±2 N). The peg and hole were initially kept in the two-point contact. Then, different TCPs were adopted to simulate different locations of the compliance centre.

## 5.2. Experimental results

The parameters for the experiments are shown in table 1. Figure 2 gives the meanings of the geometrical parameters in table 1. In addition, the stiffness along the extraction direction and the extraction speed are set at $K_z = 5\,\mathrm{N\,mm^{-1}}$ and $v = 0.01\,\mathrm{m\,s^{-1}}$, respectively. The effects of the location of the compliance centre on how extraction forces change with depth are shown in figure 16. It can be seen that the contact states were significantly different when the compliance centre was located at different positions, both in theory and in practice.

A high extraction force is a sign of two-point contact. When the compliance centre was far from the tip of the peg, the two-point contact region was the largest, evidenced by the high and wide peak of the red curve. With a decrease in $L_C$, the two-point contact region was reduced as shown by the narrower and lower peaks of the blue and black curves. These experimental results confirm the theoretical predictions of figures 12 and 13.

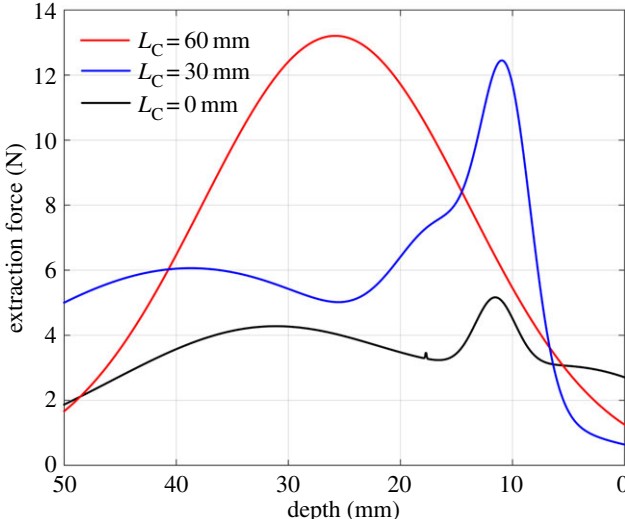

**Figure 16.** Dependence of extraction forces on the location of compliance centre. Note: force measurements were filtered using Gaussian fittings to remove noise before being plotted.

# 6. Conclusion

Disassembly is the first operation in a remanufacturing chain. The removal of a peg (shaft) from a hole (bore) is a common disassembly operation. This paper is the first to investigate that operation in depth. The work was part of a programme of research into understanding the mechanics of disassembly in order to gain an insight into the design of robotic disassembly systems.

A quasi-static analysis of peg–hole disassembly with a compliant manipulator, implementing different locations of compliance centre, initial position errors and stiffness, has been performed. The boundaries of the two-point contact region have been derived to explore the optimum position of the compliance centre. It was also found that the two-point contact region varied not only with the location of the compliance centre, but also with the initial lateral and angular errors and the lateral and angular stiffness of the manipulator. Based on the theoretical results, different movement regimes with different contact states have been distinguished corresponding to different initial conditions and compliance parameters.

The effects of the location of the compliance centre on extraction forces, as well as the contact states, were experimentally investigated. When the compliance centre was located near or at the tip of the peg, the two-point contact region and the extraction forces were small, under the same initial conditions including initial position errors and stiffness. The experimental results agreed with the theoretical model with respect to peg–hole disassembly. The experiment could be improved by adopting more accurate methods of measuring and accounting for errors in positioning the peg and manipulating the peg and the hole.

Data accessibility. All data sources are provided in the electronic supplementary material.

Authors' contributions. Y.Z., H.L. and Y.W. contributed to the quasi-static model of peg–hole disassembly and drafted the manuscript. Y.Z. contributed to the analysis of the key factors on the peg–hole disassembly and programmed the disassembly model. D.T.P. supervised the research, provided the comments on the disassembly model and experimental results and contributed to the manuscript writing. M.Q. and L.J. contributed to the experimental work of the peg–hole disassembly process. S.S. participated in the analysis of data. All authors approved the final version of the paper.

Competing interests. We declare that we have no competing interests.

Funding. This work was supported by the National Natural Science Foundation of China [grant no. 51675393]; Engineering and Physical Sciences Research Council (EPSRC) [grant no. EP/N018524/1]; the Royal Society [grant no. IEC\NSFC\181018]; the Special Fund for Key Project of Science and Technology of Hubei Province [grant no. 2017AAA111]; the Fundamental Research Funds for the Central Universities [grant no. 2016-YB-021]; and the Chinese Government Scholarship by the China Scholarship Council [grant no. 201706950051].

# Appendix A

notation

| | |
|---|---|
| $c$ | clearance ratio |
| $D$ | diameter of the hole (mm) |
| $F_x$ | lateral force (N) |
| $F_{ff1}$ | friction force at contact point 1 (N) |
| $F_{ff2}$ | friction force at contact point 2 (N) |
| $F_{N1}$ | reaction force at contact point 1 (N) |
| $F_{N2}$ | reaction force at contact point 2 (N) |
| $F_z$ | extraction force (N) |
| $h$ | extraction depth (mm) |
| $h_0$ | initial depth of the peg in the hole (mm) |
| $H$ | height of the peg (mm) |
| $K_x$ | lateral stiffness of the compliant manipulator (N mm$^{-1}$) |
| $K_z$ | vertical stiffness of the compliant manipulator (N mm$^{-1}$) |
| $K_\theta$ | rotational stiffness of the compliant manipulator (Nmm rad$^{-1}$) |
| $L_C$ | location of the compliance centre (mm) |
| $M$ | moment applied on the peg (Nmm) |
| $r$ | radius of the peg (mm) |
| $R$ | radius of the hole (mm) |
| $U_0$ | initial distance between the compliance centre and the hole axis (mm) |
| $U$ | distance between the compliance centre and the hole axis (mm) |
| $\beta_0$ | initial angular error (rad) |
| $\delta_0$ | initial lateral error (mm) |
| $\varepsilon$ | distance between the peg tip and the hole axis (mm) |
| $\varepsilon_0$ | initial distance between the peg tip and the hole axis (mm) |
| $\theta$ | tilt angle of the peg (rad) |
| $\theta_0$ | initial tilt angle of the peg (rad) |
| $\mu$ | coefficient of friction |

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
