## [Reviewer comments · Royal Society Open Science]

Review History

RSOS-190476.R0 (Original submission)

Review form: Reviewer 1

Is the manuscript scientifically sound in its present form?

Yes

Are the interpretations and conclusions justified by the results?

Yes

Is the language acceptable?

Yes

Is it clear how to access all supporting data?

Yes

Do you have any ethical concerns with this paper?

No

Have you any concerns about statistical analyses in this paper?

Yes

Recommendation?

Accept with minor revision (please list in comments)

Comments to the Author(s)

Paper is well written and makes a contribution to the state of the art. The figures are clear and well explained in the text of the paper. Development of the governing equations and their application to specific scenarios is also well done.

However the actual robotic comparison could be improved and stands as a lost opportunity.

Note:

- the 1N resolution is sufficient from a gage R&R point of view to achieve the required results.
- What is the effect of the .1 mm error on the results?
- do the curves in figure 17 have a smoothing function?
- how are these experimental results compared to the calculated values? "there was a good agreement between theoretical and experimental results." this needs to be shown and further explained.
- what are the conclusions from the experimental part of the paper? Should this be left out of the paper until more definitive results have been achieved.

Look forward to seeing a revision.

Review form: Reviewer 2

Is the manuscript scientifically sound in its present form?

Yes

Are the interpretations and conclusions justified by the results?

Yes

Is the language acceptable?

Yes

Is it clear how to access all supporting data?

Yes

Do you have any ethical concerns with this paper?

No

Have you any concerns about statistical analyses in this paper?

No

Recommendation?

Accept with minor revision (please list in comments)

Comments to the Author(s)

1. Further details of the experimental design needs to be included in section 5.1, in particular how to confirm that method to ensure the same initial lateral error and angular error in the experimental setup is accurate.
2. What were the limitations of the experimental design?

3. Page 23 states that "Although inaccuracies in the current experiment had no impact on the trends of the results" - this needs to be clarified and if there could be implications for other similar experiments but with different variables?
4. It is important to discuss the main contributions of this paper, and how this has an influence on control for remanufacturing.
5. Some papers are rather old, such as reference no 7 from 1994? (page 24); or page 25 from 1979?

Review form: Reviewer 3

Is the manuscript scientifically sound in its present form?

Yes

Are the interpretations and conclusions justified by the results?

Yes

Is the language acceptable?

Yes

Is it clear how to access all supporting data?

Yes

Do you have any ethical concerns with this paper?

No

Have you any concerns about statistical analyses in this paper?

I do not feel qualified to assess the statistics

Recommendation?

Accept with minor revision (please list in comments)

Comments to the Author(s)

Following are minor questions I have during the reading of the manuscript:

1. The authors are not taking into account the geometric tolerances that may be present during any manufacturing process. How will the presence of tolerances affect their analysis? Will this be just a static shift taken into consideration, or will it require a different approach to the problem?
2. How does the current analysis consider the presence of friction at the contact point(s) between the peg and hole? What role will it play in governing the lateral or angular stiffness of the compliance mechanism, if that is a factor?
3. During experimentation, how repeatable are the results when considering errors in positioning the peg and also in manipulating the peg within the hole? Can the authors provide possibilities in reducing the variability due to these factors when performing experiments?
4. This might be a minor detail: when the compliance center is located near the tip of the peg, for two-point contact, are there conditions under which the entire process becomes self-locking?

Decision letter (RSOS-190476.R0)

17-Jul-2019

Dear Dr Wang,

On behalf of the Editors, I am pleased to inform you that your Manuscript RSOS-190476 entitled "Peg-hole disassembly using active compliance" has been accepted for publication in Royal Society Open Science subject to minor revision in accordance with the referee suggestions. Please find the referees' comments at the end of this email.

The reviewers and handling editors have recommended publication, but also suggest some minor revisions to your manuscript. Therefore, I invite you to respond to the comments and revise your manuscript.

- Ethics statement

- Data accessibility

<http://datadryad.org/submit?journalID=RSOS&manu=RSOS-190476>

- Competing interests

- Authors' contributions

- Acknowledgements

- Funding statement

Because the schedule for publication is very tight, it is a condition of publication that you submit the revised version of your manuscript before 26-Jul-2019. Please note that the revision deadline will expire at 00.00am on this date. If you do not think you will be able to meet this date please let me know immediately.

Best regards,

on behalf of Dr Manoj Srinivasan (Associate Editor) and R. Kerry Rowe (Subject Editor)
openscience@royalsociety.org

Associate Editor Comments to Author (Dr Manoj Srinivasan):

Comments to the Author:

All three reviewers appreciated the study and have mostly minor suggestions for review. Please revise and resubmit accordingly, addressing any reviewer comments via appropriate additional sentences, comparisons, or caveats limiting the article's scope better.

Reviewer comments to Author:

Reviewer: 1

Comments to the Author(s)

Paper is well written a makes a contribution to the state of the art. The figures are clear and well explained in the text of the paper. Development of the governing equations and their application to specific scenarios is also well done.

However the actual robotic comparison could be improved and stands as a lost opportunity.

Note:

- the 1N resolution is sufficient from a gage R&R point of view to achieve the required results.
- What is the effect of the .1 mm error on the results?
- do the curved in figure 17 have a smoothing function?

- how are these experimental results compared to the calculated values? "here was a good agreement between theoretical and experimental results." this needs to be show and further explained.
- what are the conclusions from the experimental part of the paper? Should this be left out of the paper until more definitive results have been achieved.

Look forward to seeing a revision.

Reviewer: 2

Comments to the Author(s)

1. Further details of the experimental design needs to be included in section 5.1, in particular how to confirm that method to ensure the same initial lateral error and angular error in the experimental setup is accurate.
2. What were the limitations of the experimental design?
3. Page 23 states that "Although inaccuracies in the current experiment had no impact on the trends of the results" - this needs to be clarified and if there could be implications for other similar experiments but with different variables?
4. It is important to discuss the main contributions of this paper, and how this has an influence on control for remanufacturing.
5. Some papers are rather old, such as reference no 7 from 1994? (page 24); or page 25 from 1979?

Reviewer: 3

Comments to the Author(s)

Following are minor questions I have during the reading of the manuscript:

1. The authors are not taking into account the geometric tolerances that may be present during any manufacturing process. How will the presence of tolerances affect their analysis? Will this be just a static shift taken into consideration, or will it require a different approach to the problem?
2. How does the current analysis consider the presence of friction at the contact point(s) between the peg and hole? What role will it play in governing the lateral or angular stiffness of the compliance mechanism, if that is a factor?
3. During experimentation, how repeatable are the results when considering errors in positioning the peg and also in manipulating the peg within the hole? Can the authors provide possibilities in reducing the variability due to these factors when performing experiments?
4. This might be a minor detail: when the compliance center is located near the tip of the peg, for two-point contact, are there conditions under which the entire process becomes self-locking?

Author's Response to Decision Letter for (RSOS-190476.R0)

See Appendix A.

Decision letter (RSOS-190476.R1)

30-Jul-2019

Dear Dr Wang,

I am pleased to inform you that your manuscript entitled "Peg-hole disassembly using active compliance" is now accepted for publication in Royal Society Open Science.

on behalf of Dr Manoj Srinivasan (Associate Editor) and R. Kerry Rowe (Subject Editor)
openscience@royalsociety.org

Appendix A

Original Manuscript ID: ROS-190476

Original Article Title: “Peg-hole disassembly using active compliance”

To: Royal Society Open Science Editor

Re: Response to reviewers

Dear Editor,

Thank you for your letter.

We have addressed all questions from the reviewers in the revised manuscript, as attached.

Best regards,

Corresponding author:
Dr. Yongjing Wang
Department of Mechanical Engineering
School of Engineering
University of Birmingham
B15 2TT
United Kingdom
E-mail: Y.Wang@bham.ac.uk

Reviewer#1:

Paper is well written and makes a contribution to the state of the art. The figures are clear and well explained in the text of the paper. Development of the governing equations and their application to specific scenarios is also well done.

Reviewer#1, Comment # 1: The 1N resolution is sufficient from a gage R&R point of view to achieve the required results. What is the effect of the .1 mm error on the results?

Author response:

The experiment was repeated at least three times; the consistency of the results proved that the effects of the repeatability of the robots were negligible.

Author action:

We have revised the manuscript. Please see the yellow highlighting in Section 5.1.

Reviewer#1, Comment # 2: Do the curves in figure 17 have a smoothing function?

Author response:

Yes, the curves shown in the figure are the processed results from the original experimental data by the polynomial fitting method.

Author action:

We have added a note to Figure 16 (Figure 17 in the old version).

Reviewer#1, Comment # 3: How are these experimental results compared to the calculated values? "there was a good agreement between theoretical and experimental results." this needs to be shown and further explained.

Author response:

A high extraction force is a sign of two-point contact. When the compliance centre was far from the tip of the peg, the two-point contact region was the largest, evidenced by the high and wide peak of the red curve. With a decrease in L_c , the two-point contact region was reduced as shown by the narrower and lower peaks of the blue and black curves. These experimental results confirm the theoretical predictions of Figs. 12 and 13.

Author action:

We have revised the manuscript. Please see the yellow highlighting in Section 5.2.

Reviewer#1, Comment # 4: What are the conclusions from the experimental part of the paper? Should this be left out of the paper until more definitive results have been achieved.

Author response:

As shown above, the experiment results agree well with the theoretical prediction. We are keeping the experimental part of the paper to show that the theoretical analysis has been experimentally validated.

Reviewer#2, Comment # 1: Further details of the experimental design needs to be included in section 5.1, in particular how to confirm that method to ensure the same initial lateral error and angular error in the experimental setup is accurate.

Author response:

The robot was programmed to move to the same start position before beginning to pull the peg out of the hole. The start position was chosen to induce the desired contact forces on the peg. It was found that the position repeatability of the robot ($\pm 0.1\text{mm}$) was sufficient to ensure the initial contact forces were within the required tolerance ($\pm 2\text{ N}$).

Author action:

We have revised the manuscript. Please see the blue highlighting in Section 5.1.

Reviewer#2, Comment # 2: What were the limitations of the experimental design?

Author response:

The limitation was the repeatability of the robot and force measurements which we have now specified.

Reviewer#2, Comment # 3: Page 23 states that "Although inaccuracies in the current experiment had no impact on the trends of the results" - this needs to be clarified and if there could be implications for other similar experiments but with different variables?

Author response:

This statement refers to the difficulty of measuring lateral and angular errors. If they can be measured accurately, more analysis can be carried out. The absence of such analysis does not affect the current conclusion of the paper.

Author action:

To avoid confusion, we have amended the text slightly.

Reviewer#2, Comment # 4: It is important to discuss the main contributions of this paper, and how this has an influence on control for remanufacturing.

Author action:

We have revised Conclusion. Please see the yellow highlighting in Conclusion.

Reviewer#2, Comment # 5: Some papers are rather old, such as reference no 7 from 1994? (page 24); or page 25 from 1979?

Author response:

The old references represent key developments of using RCC to assist robotic assembly. We adopted a similar approach in the study of robotic disassembly. To our knowledge, there are no papers on disassembly mechanics so far.

Reviewer#3:

Following are minor questions I have during the reading of the manuscript:

Reviewer#3, Comment # 1: The authors are not taking into account the geometric tolerances that may be present during any manufacturing process. How will the presence of tolerances affect their analysis? Will this be just a static shift taken into consideration, or will it require a different approach to the problem?

Author response:

The errors in diameters caused by manufacturing processes can be reflected by the geometric tolerance which has been considered in the model as shown in Section 3.1. If the manufacturing processes cause the shape of the pin or the hole to change, the removal process should be modelled differently and this is beyond the scope of this research.

Reviewer#3, Comment # 2: How does the current analysis consider the presence of friction at the contact point(s) between the peg and hole? What role will it play in governing the lateral or angular stiffness of the compliance mechanism, if that is a factor?

Author response:

We used Coulomb friction in the modelling, as shown in Equations 4-6 in Section 3.1, which has been applied to a series of previous research on assembly mechanics. Determining the stiffness of compliance is an independent topic and beyond the scope of this paper.

Reviewer#3, Comment # 3: During experimentation, how repeatable are the results when considering errors in positioning the peg and also in manipulating the peg within the hole? Can the authors provide possibilities in reducing the variability due to these factors when performing experiments?

Author response:

It was found that the position repeatability of the robot ($\pm 0.1\text{mm}$) was sufficient to ensure the initial contact forces were within the required tolerance ($\pm 2\text{ N}$).

The experiment was repeated at least three times; the consistency of the results proved that the effects of the repeatability of the robots were negligible.

Author action:

We have revised the manuscript. Please see the highlighting parts in Sections 5.1 and 5.2.

Reviewer#3, Comment # 4: This might be a minor detail: when the compliance center is located near the tip of the peg, for two-point contact, are there conditions under which the entire process becomes self-locking?

Author response:

The purpose of locating the compliance centre near the tip of the peg is to reduce the likelihood of self-locking as shown in Figs. 12 and 13 in the paper (and also, in the context of assembly, in the work of Whitney etc.)